# Xanthohumol and Gossypol Are Promising Inhibitors against *Babesia microti* by In Vitro Culture via High-Throughput Screening of 133 Natural Products

**DOI:** 10.3390/vaccines8040613

**Published:** 2020-10-16

**Authors:** Jiaying Guo, Xiaoying Luo, Sen Wang, Lan He, Junlong Zhao

**Affiliations:** 1State Key Laboratory of Agricultural Microbiology, College of Veterinary Medicine, Huazhong Agricultural University, Wuhan 430070, China; jiaying@webmail.hzau.edu.cn (J.G.); senwang@webmail.hzau.edu.cn (S.W.); zhaojunlong@mail.hzau.edu.cn (J.Z.); 2Key Laboratory of Preventive Veterinary Medicine in Hubei Province, Wuhan 430070, China; 3College of Animal Sciences, Fujian Agriculture and Forestry University, Fuzhou 350000, China; luoxiaoying@fafu.edu.cn; 4Key Laboratory of Animal Epidemical Disease and Infectious Zoonoses, Ministry of Agriculture, Huazhong Agricultural University, Wuhan 430070, China

**Keywords:** human babesiosis, *Babesia microti*, natural products, gossypol, xanthohumol

## Abstract

Human babesiosis caused by *Babesia microti* is an emerging threat for severe illness and even death, with an increasing impact worldwide. Currently, the regimen of atovaquone and azithromycin is considered as the standard therapy for treating human babesiosis, which, however, may result in drug resistance and relapse, suggesting the necessity of developing new drugs to control *B. microti*. In this regard, natural products are promising candidates for drug design against *B. microti* due to their active therapeutic efficacy, lower toxicity, and fewer adverse reactions to host. Here, the potential inhibitors against *B. microti* were preliminarily screened from 133 natural products, and 47 of them were selected for further screening. Gossypol (Gp) and xanthohumol (Xn) were finally shown to effectively inhibit the growth of *B. microti* with IC_50_ values of 8.47 μm and 21.40 μm, respectively. The cytotoxicity results showed that Gp and Xn were non-toxic to erythrocytes at a concentration below 100 μm. Furthermore, both of them were confirmed to be non-toxic to different types of cells in previous studies. Our findings suggest the potential of Gp and Xn as effective drugs against *B. microti* infection.

## 1. Introduction

Babesiosis is an emerging zoonosis induced by the intraerythrocytic protozoa of the genus *Babesia*, which may cause enormous harm, threat, and loss to the livestock industry and even public health [1,2,3]. Facing these challenges, the Center for Disease Control and Prevention regarded it as a nationally notifiable disease and has been monitoring it in the United States since January 2011 [4,5]. To date, more than 100 *Babesia* species have been reported, but only a few of them can infect humans and cause human babesiosis, including *Babesia microti*, *Babesia divergens*, *Babesia duncani,* and *Babesia venatorum* [2,6]. *B. microti* is the primary agent for human babesiosis and can be transmitted by tick *Ixodes scapularis*, blood transfusion, and placenta [2,7]. Babesiosis cases are distributed throughout the world, especially in the United States, Asia, and Europe, and show an upward trend worldwide [8,9,10]. The clinical manifestations are similar to those of malaria, including fever, chills, hemolysis, hemoglobinuria, splenomegaly, and jaundice [4]. The symptoms for the patients infected with *B. microti* depend on their age and physical health status. The young and those in good health are always asymptomatic, whereas the elderly, neonates and those with deficient immunity tend to be have more serious symptoms, with a mortality rate over 20% for patients with malignant tumors, splenectomy, hemoglobinopathy, and human immunodeficiency virus (HIV) infections [2].

Despite some recommended therapies available for *Babesia*, some challenges still need to be solved. So far, imidocarb dipropionate (ID) and diminazene aceturate (DA) are the two main babesiacides and are usually used as the most effective anti-*Babesia* drugs [11,12]. However, drug resistance has become a limitation to the efficiency of DA, leading to its restriction in Europe and difficulty expanding it globally [11,13]. ID is not available all around the world due to the problem of drug residue and toxicity [14,15]. An increasing number of limits have been imposed on the use of single-component drugs, in contrast to the increasing use of drug combinations in anti-*Babesia*. The combination of clindamycin with quinine has been reported as a conventional regimen especially effective in the treatment of moderate to severe *Babesia*-infection based on clinical trials and experiments with hamsters [16]. However, half of the patients who took clindamycin and quinine showed adverse effects, such as vertigo, tinnitus, and gastroenteritis. Based on the failure and adverse effects in response to clindamycin and quinine, atovaquone, an analogue of quinine, was used in combination with azithromycin as an alternative therapy against *Babesia* infection according to its efficacy against malaria [17,18]. The regimen of atovaquone and azithromycin is effective not only in *B. microti-*infected hamsters, but also in the patients who have failed with clindamycin and quinine [18]. The alternative therapy plays a major role in asymptomatic or mildly symptomatic patients and has been reported to be applied even in severe *Babesia* infections. Moreover, the therapy of atovaquone and azithromycin was shown to have fewer adverse effects than that of clindamycin and quinine [19]. These reports indicate the necessity to develop new treatment therapies against *B. microti* infection due to the extensive problems in the combination of clindamycin with quinine, such as drug resistance, adverse effects, drug residues, relapse, and treatment failure.

The cases of patients infected with *B. microti* are on the rise annually worldwide, and the transmission of *B. microti* through blood transfusion greatly increases the risk of infection [7]. To date, there are no effective vaccines against *B. microti* infection [1]. Although some existing drugs can play a role in inhibiting *Babesia*, there are still many problems, such as the efficacy of the drugs and their adverse effects on patients. Moreover, drug resistance is a growing problem that cannot be neglected.

Natural products, the substances derived from plants, animals, marine creatures, and microbes, may help to solve these problems [20,21]. Due to their intricate molecular frameworks and lack of effective extraction methods, the development of natural product-derived drugs is slow when compared with that of chemically synthesized drugs [22,23]. However, natural products have several advantages over synthetic compounds, such as higher number of chiral centers, steric complexity, oxygen atoms, and a wider distribution of molecular properties [22]. Because of the available interactions with many specific targets as well as lower toxicity and fewer adverse reactions to the host, natural products are perceived as valuable sources for drug discovery and potential clinical trial candidates [24]. To date, natural products have been extensively applied in many diseases, such as infectious diseases, immunological diseases, tumor diseases, and parasitic diseases [20,25,26,27]. Artemisinin and its derivatives (dihydroartemisinin, artesunate, arteether, and artemether), extracted and separated from Artemisia annua, play significant roles in anti-malaria and are considered as the most important drugs against malaria [28]. Moreover, an increasing number of studies focus on the role of artemisinin in combating *Babesia* infection [28].

Against the above background, the aim of the present study was to screen out potential and effective alternative anti-*Babesia* drugs from natural products and find an effective method to inhibit *B. microti* infection.

## 2. Materials and Methods

### 2.1. Ethics Statement

All the experimental animals were housed and treated in accordance with the stipulated rules for the regulation of the administration of affairs concerning experimental animals of the People’s Republic of China. All the experiments were performed under the approval of Laboratory Animals Research Centre of Hubei Province and the Ethics Committee of Huazhong Agricultural University (Permit number: HZAUMO-2019-005).

### 2.2. Parasites and Mice

*B. microti* (ATCCR PRA-99™) was provided by the National Institute of Parasitic Diseases, Chinese Center for Disease Control and Prevention (Shanghai, China). The strain of *B. microti* was maintained by passage in the blood of Kunming (KM) mice in our laboratory (State Key Laboratory of Agricultural Microbiology, College of Veterinary Medicine, Huazhong Agricultural University, Wuhan, Hubei, China).

### 2.3. Preparation of Infected Erythrocytes

The parasitemia of *B. microti*-infected KM mice was microscopically determined on blood smears. When the parasitemia reached 15–20%, 100 μL *B. microti*-infected blood was collected from each mouse for further analysis. Briefly, mouse blood was collected into sterile vacuum tubes containing anticoagulant with EDTA (100 μL/mL blood), followed by centrifugation at 3000 rpm for 5 min at 4 °C. Next, the supernatant, especially the layer of white blood cells, was removed and the pellet was resuspended with cold Puck’s saline glucose buffer (PSG) to the original volume and then gently mixed. The above step was repeated three times or more until the supernatant was clear. Finally, the resulting mixture was resuspended 1:1 with Puck’s saline glucose plus extra glucose (PSG + G) and stored at 4 °C for further analysis.

### 2.4. In Vitro Culture of B. microti

A 96-well flat-bottom plate was used for *B. microti* culture and drug screening. The culture conditions consisted of 25 μL of infected red blood cells (RBCs), 15 μL of uninfected normal RBCs (10% hematocrit), 110 µL of culture medium supplemented with 2% HB-101 (Irvine Scientific, Shanghai, China), 20% fetal bovine serum (FBS, ATLANTA Biologicals, Shanghai, China), 10 mg/L Albumax I (Gibco Life Technologies, Shanghai, China), 2 mm L-glutamine (Irvine Scientific, Shanghai, China), 2% Antibiotic/Antimycotic 100× (Corning, Shanghai, China), and hypoxanthine (200 µm)-thymidine (30 µm) (Sigma-Aldrich, St. Louis, MO, USA) [29]. The culture plate was incubated at 37 °C with a gas mixture of 2% O_2_, 5% CO_2_, and 93% N_2_ at a constant pressure in a modular incubator chamber (Billups-Rothenberg, Del Mar, CA, USA). The culture was split every three days and the medium was replaced with HL-20 every 24 h as a routine management until 72 h as follows: 110 μL supernatant was discarded and 0.3 μL of the culture was drawn from the bottom for parasitemia determination, followed by adding 110 μL HL-20 medium into the well and gently mixing. The thin smears were fixed with ice cold 50% ethanol and 50% methanol (v:v) and stained with Giemsa (Thermo Fisher Scientific, Cleveland, OH, USA). Finally, the percentage of parasitized erythrocytes was determined by microscopy and flow cytometry.

### 2.5. Preliminary Screening and Rescreening of Natural Products

A total of 133 natural products (different herbal extracts) obtained from the small molecule natural product library (Selleck Chemicals, Houston, TX, USA) were used for screening potential drugs against *B. microti* by in vitro culture. The general information of the 133 natural products is shown in Appendix A. The assays were performed in triplicate and in 96-well flat-bottom plates. For initial screening, 133 drugs were diluted separately by HL-20 medium to the final concentration of 10 μm (50 μm for rescreening). Cells treated with DMSO (0.01%) alone were used as negative controls. DA is a recommended drug reported to be effective in inhibiting *B. microti* and it was used as a positive control in this study. During the experiment, *B. microti* was cultured under the drug-free condition for 24 h and the parasitemia was determined as the initial parasitemia. Next, 110 μL supernatant was replaced with the HL-20 medium that contained drugs. The parasitemia was determined separately at 24 h, 48 h, and 72 h by microscopy and flow cytometry. The 1 mg/mL Hoechest-33258 (Beyotime Biotechnology, Shanghai, China) was diluted by phosphate buffer saline (PBS) that had passed through 0.22 μm filters to 100 μg/mL. Next, 500 μL PBS, 10 μL Hoechest-33258 (100 μg/mL), and 30 μL erythrocyte pellet collected from 96-well plates were mixed gently in 1.5 mL tubes and were incubated in the dark at 37 °C for 30 min. Finally, the parasitemia was determined by flow cytometry and calculated for every sample based on 100,000 cells. The data were analyzed by GraphPad Prism (version 6.0, La Jolla, CA, USA) (with significant differences indicated as **** *p* < 0.0001, *** *p* < 0.001, ** *p* < 0.01, and * *p* < 0.05). The parasitemia at 72 h was compared with the parasitemia of negative and positive controls. The drugs with significant differences from DMSO and no significant differences from DA in efficacy were selected for rescreening.

### 2.6. Combination of Gp and Xn on In Vitro Culture

Gp and Xn were combined to determine whether their effects are additive, synergistic, or antagonistic. The combination assay was performed using the Chou–Talalay method at the concentrations of 0.25 × IC_50_, 0.5 × IC_50_, IC_50_, 2 × IC_50_, and 4 × IC_50_ (0 μm, 2.11 μm, 4.23 μm, 8.47 μm, 16.94 μm, and 33.88 μm for Gp and 0 μm, 5.35 μm, 10.7 μm, 21.4 μm, 42.8 μm, and 85.6 μm for Xn). Different concentration combinations were diluted by HL-20 medium in 96-well flat-bottom plates in triplicate. DA at 1 μm was used as the positive control. The culture plate was incubated with a mixture gas of 2% O_2_, 5% CO_2_, and 93% N_2_ at 37 °C at a constant pressure. After 72 h of incubation, 0.6 μL of the culture was collected from the bottom to make smears for parasitemia determination. The parasitemia data for the combination assay were analyzed using the CompuSyn software (ComboSyn, Inc., Paramus, NJ, USA) based on the Chou–Talalay method [30].

### 2.7. Cell Toxicity Assay

Gp and Xn were added into the HL-20 medium at the final concentrations of 1 μm, 10 μm, 50 μm, and 100 μm, respectively. The culture conditions for the cytotoxicity assay consisted of 110 μL HL-20 medium that contained different concentrations of drugs and 40 μL infected mouse RBCs, which were mixed in the 96-well plates in triplicate. The culture medium was changed every 24 h to 72 h, from which 10 μL was collected for blood count. Further significance tests were performed by GraphPad Prism 6 (Graphpad Software, La Jolla, CA, USA).

### 2.8. In Vivo Inhibition Assay

KM mice (six weeks) were given intraperitoneal injection of *B. microti* that had been preserved in liquid nitrogen. A drop of blood was collected from the tail vein for Giemsa staining to determine the parasitemia after 24 h of injection. On the fifth day, the KM mice were injected intramuscularly and separately with DMSO (3 mg/kg), DA (3 mg/kg), Gp, and Xn in triplicate. The injection concentration gradient was 1 mg/kg, 3 mg/kg, and 5 mg/kg for Gp and 1 mg/kg, 5 mg/kg, and 10 mg/kg for Xn. Parasitemia was determined by microscopy.

## 3. Results

### 3.1. Forty-Seven Natural Products Showed Significant Inhibitory Effects on B. microti in Preliminary Screening

The inhibitory effects of 133 natural products (10 μm) on *B. microti* were explored by in vitro culture, and 47 of them were selected for rescreening through comparison with DMSO and DA (Figure 1 and Appendix A). For comparison with DMSO, the inhibitory effects were calculated using the formula: ((parasitemia for DMSO minus parasitemia for natural products)/parasitemia for DMSO) × 100% (Figure 2 and Appendix A). Significance analysis was carried out and the natural products with no significant difference from DA were selected for rescreening. A comparison with DA and DMSO further selected 47 natural products with the code numbers of A7, A8, B4, B5, B8, B9, C10, E5, E9, E10, E11, G3, G4, G5, G8, H2, H4, H7, H8, 2A1, 2A4, 2A5, 2B1, 2B4, 2B6, 2C3, 2D1, 2D2, 2D3, 2D4, 2D5, 2D6, 2E2, 2E3, 2E5, 2E6, 2F1, 2F2, 2F3, 2F5, 2G2, 2G4, 2G5, 2H1, 2H2, 2H4, and 2H5. The full names of the 47 natural products are detailed in Appendix A.

The inhibitory effects of 47 natural products on *B. microti* were rescreened under 50 μm (Figure 3). Among them, 2F1 (dioscin) was discarded due to the dissolution of RBCs. After 72 h in vitro culture, the data were subjected to significance analysis, and the parasitemia for five of the natural products was shown to be not significantly different from that of DA: 2E6 (Xn), G3 (cryptotanshinone), G5 (honokiol), 2F3 (palmatine chloride), and B5 (Gp) with inhibition rates of 66.97%, 47.58%, 51.50%, 48.64%, and 63.05%, respectively (Figure 4). The chemical structure and formula of the five natural products obtained from the small molecule natural product library (Selleck Chemicals, Houston, TX, USA) are shown in Appendix A.

To confirm the results of parasitemia by microscopy, parasitemia under the treatment of 47 natural products for 72 h was also determined by flow cytometry. Hoechst was used as a marker for *B. microti* detection and the parasitemia was calculated by the percent of infected RBCs in 100,000 RBCs. A comparison with DMSO showed that *B. microti* was significantly inhibited by 2E6 (Xn) and B5 (Gp) with inhibition rates of 64.53% and 53.20%, respectively (Figure 5). Meanwhile, the inhibition rates for 2E6 and B5 showed no significant differences from that of DA (63.99%). The other three of the five natural products, 2F3, G3, and G5 (palmatine chloride, cryptotanshinone, and honokiol), were excluded due to no significant difference between their inhibitory effect and that of DMSO by flow cytometry determination. Finally, only Xn and Gp were identified to have significant inhibitory effects on *B. microti* and thus selected for further analysis.

### 3.2. Cytotoxicity of Gp and Xn

After treatment with 1 μm, 10 μm, 50 μm, and 100 μm Gp for 72 h, the values of RBC counts were 1.44 × 10^9^/mL, 1.46 × 10^9^/mL, 1.22 × 10^9^/mL, and 1.04 × 10^9^/mL, respectively. For 1 μm, 10 μm, 50 μm, and 100 μm Xn, the values of RBC counts were 1.38 × 10^9^/mL, 1.41 × 10^9^/mL, 1.40 × 10^9^/mL, and 1.34 × 10^9^/mL, respectively. All the values of RBC counts showed no significant difference from the negative control for all the concentration gradients of Xn (Figure 6A) and Gp (Figure 6B) according to the statistical analysis. These results indicated that Gp and Xn were non-toxic to RBCs below 100 μm and their inhibitory effects on *B. microti* could be further evaluated. The cytotoxicity of Gp and Xn on other cells and organisms have been extensively investigated in previous studies [31,32,33,34]. Based on the results of cytotoxicity of Xn and Gp on erythrocytes and the reports of their cytotoxicity on other cells, Xn and Gp are non-toxic at IC_50_ values of 21.40 μm and 8.47 μm, respectively, and they can be considered as effective inhibitor candidates against *B. microti* (Table 1).

### 3.3. The Inhibitory Effect of Xn and Gp on the In Vitro Culture of B. microti

After treating *B. microti* for 72 h with 1 μm, 10 μm, and 50 μm of Gp and Xn, the parasitemia was 4.20%, 2.07%, and 1.93% for Gp and 3.87%, 3.93%, and 2.07% for Xn, respectively. The results of Gp and Xn showed significant difference from that of DMSO (*p* < 0.0001), but no significant difference from that of DA (*p* > 0.05). Meanwhile, 10 μm Gp and 50 μm Xn were shown to extensively inhibit the reproduction of *B. microti*. Based on the parasitemia at 72 h, the inhibitory effects of Gp (Figure 7A,B) and Xn (Figure 7C,D) on the in vitro culture of *B. microti* were evaluated, and the IC_50_ values for Gp and Xn were 8.47 μm and 21.40 μm, respectively, according to the significance analysis.

### 3.4. The Inhibitory Effect of Xn and Gp Combination on the In Vitro Culture of B. microti

Six dilutions (0.25 × IC_50_, 0.5 × IC_50_, IC_50_, 2 × IC_50_, and 4 × IC_50_) of Gp and Xn were combined, respectively. The parasitemia data were analyzed by CompuSyn software with the combination index (CI) values for synergism (less than 0.90), additive effect (from 0.90 to 1.10), or antagonism (more than 1.10). The CI values for the combination of drugs at IC_50_, IC_75_, IC_90_, and IC_95_ were 1.81893, 1.59855, 1.40495, and 1.28689, respectively, indicating the antagonism of the Gp and Xn combination on *B. microti*.

### 3.5. Inhibitory Effect of Xn and Gp on the In Vivo Culture of B. microti

At 24 h post administration of Xn and Gp, the parasitemia for the treatment with 1mg/kg, 3 mg/kg, and 5 mg/kg Gp showed slight increases from 20.57% to 24.97%, 19.33% to 21.93%, and 17.77% to 22.17% with growth rates of 21.39%, 13.45%, and 24.76%, respectively (Figure 8A). Consistent with Gp, after treatment of 1 mg/kg Xn, the parasitemia was slightly increased, in contrast to a significant decrease from 24.77% to 19.87% and 23.23% to 18.87% in the parasitemia of groups treated with 5 mg/kg and 10 mg/kg Xn, respectively (Figure 8B). The results indicated 5 mg/kg and 10 mg/kg Xn had better inhibitory effects on the in vivo culture of *B. microti* relative to that of 1mg/kg, 3 mg/kg, and 5 mg/kg Gp. The growth rate was much higher in the DMSO treatment than it was in the treatment with Gp or Xn.

## 4. Discussion

Compared with *B. divergens*, *B. duncani,* and *B. venatorum*, *B. microti-*caused human babesiosis has predominated and presented a gradual upward trend throughout the world [2,4]. To date, a series of drugs are available, and among them, ID and DA are two effective drugs widely used in the treatment of animal babesiosis [12]. For human babesiosis, the regimen of clindamycin and quinine was initially used as the standard treatment [18]. However, increasing problems emerged, particularly the adverse effects for patients, such as tinnitus, diarrhea, gastroenteritis, and decreased hearing [18]. The combination of atovaquone and azithromycin was regarded as an alternative therapy in curing human babesiosis caused by *B. microti,* due to fewer adverse effects and higher therapeutic efficacy [19]. However, under the regimen of atovaquone and azithromycin, increasing evidence indicated higher dosage, longer duration, and even treatment failure in some immunocompromised patients [35]. A possible explanation is associated with the mutations in the binding regions of the target proteins. A previous article reported that the treatment of an immunocompromised patient with atovaquone and azithromycin for six weeks showed the emergence of resistant haplotypes, leading to relapse, due to the mutation of tyrosine to cysteine amino acid substitution in CYTb (atovaquone binding region) and arginine to cysteine amino acid substitution in RPL4 (azithromycin binding region) [35]. While the combination of atovaquone and azithromycin is the most effective therapy against *B. microti* infection, drug resistance, relapse, and treatment failure are all pressing concerns, suggesting that the development of new therapies, especially new drugs, may be the most effective method to control *B. microti* infection.

Natural products, due to their broad range of activity, have been extensively studied in several diseases, especially in anti-parasite and anti-cancer therapies. In this study, natural products were considered as promising candidates for drug design against *B. microti* infection. The 133 natural products were obtained from the small molecule natural product library (Selleck Chemicals, Houston, TX, USA) and were preliminarily screened at 10 μm. Among them, 47 natural products were determined to be significantly different from DMSO, suggesting they could inhibit the replication of *B. microti*. After rescreening the 47 natural products at 50 μm, five of them were shown to have strong inhibitory effects on *B. microti* under microscopic examination, those products were 2F3 (palmatine chloride), B5 (Gp), 2E6 (Xn), G3 (cryptotanshinone), and G5 (honokiol). Palmatine chloride as a hydrochloride salt of palmatine has strong anti-malarial activity with low cytotoxicity and is also valuable as an Alzheimer’s disease modifying strategy [36,37]. Cryptotanshinone, a quinoid diterpene isolated from *Salvia miltiorrhizabunge,* has shown various anti-cancer activities [38]. Honokiol originally obtained from magnolia extract is reported to have the activity of inhibiting multiple malignancies [25]. However, palmatine chloride, cryptotanshinone, and honokiol were excluded through parasitemia determination by flow cytometry. Gp and Xn were finally identified to significantly inhibit the growth of *B. microti* by the combined determination of microscopy and flow cytometry. While microscopy is widely used in parasitemia calculation, this method has many disadvantages [3,39,40]. For instance, this method is not applicable or may consume vast resources of manpower in large-scale screening of potential drugs [41]. Moreover, the errors caused by factitious factors are inevitable and may influence the results of screening to a great extent [41]. The fluorescence-based method where SYBR green I (SG I) binds to the double-stranded DNA of the parasites has been extensively applied in anti-malaria drug screening [40,42]. Due to the absence of a nucleus in erythrocytes, the cells that were labeled by fluorescence were all the parasites-infected erythrocytes. The flow cytometry used in this study possesses the feature of quick analysis and is the most advanced cell analysis technology available to date [40]. Therefore, the flow cytometry screening eliminated three of the five drugs that were selected by microscopy. However, flow cytometry also has the disadvantage of high cost, and may cost too much in large-scale preliminary screening [40]. Therefore, these two methods were combined in the present study: microscopy examination in a preliminary large-scale screening and flow cytometry in rescreening confirmation. Collectively, microscopy combined with flow cytometry will facilitate resource utilization and parasitemia calculation.

Lactate dehydrogenase (LDH), an indispensable active enzyme, was reported to be involved in glycolysis, mediating the reversible metabolism of pyruvate to lactate [43]. The inhibition of LDH will lead to the death of parasites, which has been reported in *Plasmodium falciparum*, *Toxoplasma gondii*, and *Babesia bovis* [44,45]. Gp, extracted from cotton seeds, is a competitive inhibitor against the binding of reduced nicotinamide adenine dinucleotide to LDH (45). Therefore, Gp was regarded as a promising drug candidate in anti-tumor, anti-parasite, and anti-HIV research [44]. However, at a concentration over 500 μm, Gp was reported to be cytotoxic and accompanied with hemolysis by in vitro culture [46]. In this study, cytotoxicity assays were performed at 1 μm, 10 μm, 50 μm, and 100 μm of Gp and Xn, respectively. Both of them exhibited no toxicity to erythrocytes and thus can be used as candidates for drug design. The cytotoxicity to other cells and organisms has been extensively investigated in previous studies. Gp was shown to cause 50% death of chronic lymphocytic leukemia cells at 30 μM [47]. For epithelial cervix cancer cells (HeLa), brain malignant glioma cells (U87), and gastric cancer cells (M85), the IC_50_ values were 31.3 μm, 59.6 μm, and 39.7 μm, respectively [33]. The toxicity assay in vivo was also conducted, and the median lethal doses (LD_50_) were tested as 2315 mg/kg, 550 mg/kg, and 35 mg/kg for rat, pig, and mouse, respectively [48,49,50,51,52]. Xn is a prenylated chalcone derived from a hop plant, *Humulus lupulus* L. [53,54], which possesses a broad spectrum against virus, bacteria, fungi, and plasmodium [54]. Several studies explored the anticarcinogenic properties of Xn as a cancer chemo-preventive agent by modulating the enzymes involved in carcinogen metabolism and detoxification [55]. Xn can also eliminate reactive oxygen species, thereby inhibiting the production of nitric oxide and superoxide anion radicals [56]. The biological properties of Xn are extensive and have attracted the attention of an increasing number of researchers; however, few reports are available about the inhibitory effects of Xn on parasites. Thus far, the anti-leishmanial activity of Xn was only shown to be mediated by inhibiting mitochondrial activities [53]. To test the cytotoxicity, an MTT assay was performed by incubating HeLa, A549 (hypotriploid alveolar basal epithelial cells), and HepG2 (human liver hepatocellular carcinoma cell line) with Xn, and the IC_50_ values were determined as 40.4 μm, 30.5 μm, and 35.0 μm, respectively [31]. Furthermore, the toxicity of Xn to the HCT116 and HT29 as well as the hepatocellular carcinoma cell lines HepG2 and Huh7 was experimentally determined in previous studies, and the IC_50_ values were 40.8 μm, 50.2 μm, 25.4 μm, and 37.2 μm, respectively [32]. Additionally, female BALB/c mice were fed with 1000 mg Xn/kg body weight, and no signs of Xn-toxicity or adverse effects were observed in normal organs [57]. The treatment of female Sprague Dawley rats with 1000 mg Xn/kg body weight only showed weak hepatotoxicity and did not affect the development of the rats [34]. Xn was also used at an escalating dose in menopausal women, leading to no sex hormones, blood clotting, or acute toxicity [58]. All these results indicate that Gp and Xn with IC_50_ values of 8.47 μm and 21.40 μm, respectively, were non-toxic by in vitro and in vivo assays. In this article, the combination of Gp and Xn showed an antagonistic effect on *B. microti* in an in vitro test, indicating that monotherapy is better than combination therapy. For the combination therapy strategy, the drugs with a similar target will be more suitable, such as mycophenolate mofetil, AGI-5198, disulfiram, olutasidenib, and brequinar, which are potential dehydrogenase inhibitors and whose combination with Gp against *B. microti* needs to be further studied. Previous studies have shown that Xn could inhibit the cyclooxygenase 1 (COX-1) and COX-2 activities that are involved in mucosal protection [59]. Diclofenac sodium, lornoxicam, and ketorolac tromethamine salt are selective COX inhibitors, whose combination with Xn against *B. microti* could also be further investigated in the future.

Based on the preliminary screening and rescreening of 133 natural products, Gp and Xn were identified as two potential drugs against *B. microti*. On one hand, Gp (IC_50_ 8.47 μm) and Xn (IC_50_ 21.40 μm) can extensively inhibit the growth of *B. microti* at a low concentration. On the other hand, both of them were shown to be non-toxic to erythrocytes and other types of cells or organisms at the IC_50_ 8.47 μm of Gp or IC_50_ 21.40 μm of Xn. These results indicate their potential use for drug design against babesiosis caused by *B. microti*.

## 5. Conclusions

In this study, the inhibitory effects of 133 natural products on *B. microti* were preliminarily screened by microscopy, and 47 of them were further screened by flow cytometry. The combined screening and in vitro culture results revealed Gp and Xn (with IC_50_ values of 8.47 μm and 21.40 μm, respectively) as promising chemotherapeutic agents against *B. microti*. Moreover, the in vivo results also indicated their effective inhibition on *B. microti*. This study provides new insights for the design and development of novel drugs against *B. microti*, which may effectively inhibit *B. microti* infection.

## Figures and Tables

**Figure 1 vaccines-08-00613-f001:**
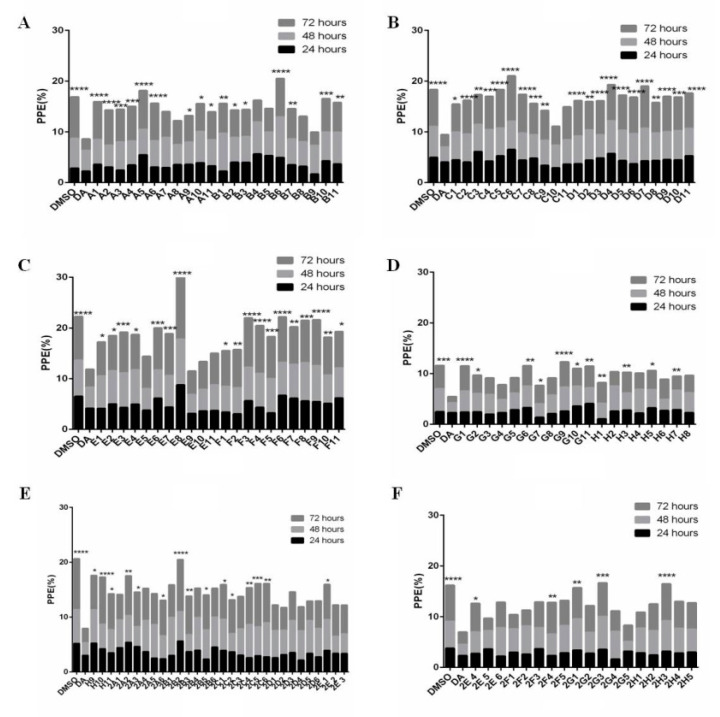
The preliminary screening of 133 natural products against *B. microti* by in vitro culture. The asterisks indicate statistically significant differences between the treated and control groups (**** *p* < 0.0001, *** *p* < 0.001, ** *p* < 0.01, * *p* < 0.05). (**A**) The natural products from A1 to B11. (**B**) The natural products from C1 to D11. (**C**) The natural products from E1 to F11. (**D**) The natural products from G1 to H8. (**E**) The natural products from H9 to 2E3. (**F**) The natural products from 2E4 to 2H5.

**Figure 2 vaccines-08-00613-f002:**
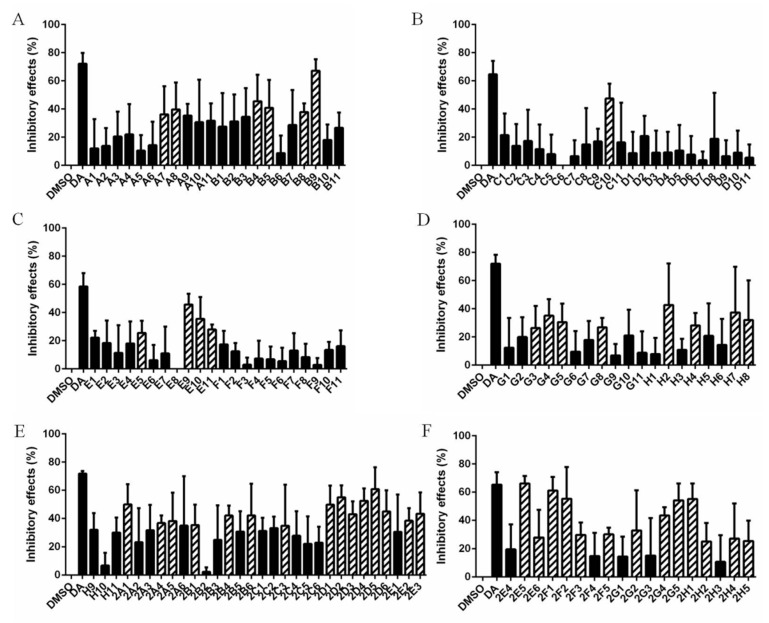
The calculated inhibitory effects of 133 natural products on *B. microti*. The error bars indicate standard error of the means for each tested group. The hollow cylinders indicate the natural products with significant difference from DMSO that were subsequently used for rescreening. (**A**) The natural products from A1 to B11. (**B**) The natural products from C1 to D11. (**C**) The natural products from E1 to F11. (**D**) The natural products from G1 to H8. (**E**) The natural products from H9 to 2E3. (**F**) The natural products from 2E4 to 2H5.

**Figure 3 vaccines-08-00613-f003:**
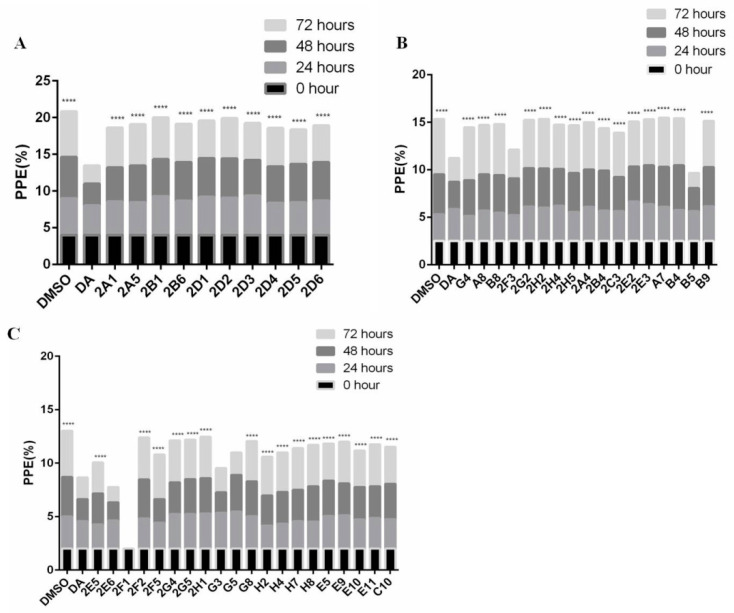
The rescreening of 47 natural products against *B. microti* by in vitro culture. The asterisks indicate statistically significant differences between the treated and control groups (**** *p* < 0.0001). (**A**) The natural products of 2A1, 2A5, 2B1, 2B6, 2D1, 2D2, 2D3, 2D4, 2D5 and 2D6. (**B**) The natural products of G4, A8, B8, 2F3, 2G2, 2H2, 2H4, 2H5, 2A4, 2B4, 2C3, 2E2, 2E3, A7, B4, B5 and B9. (**C**) The natural products of 2E5, 2E6, 2F1, 2F2, 2F5, 2G4, 2G5, 2H1, G3, G5, G8, H2, H4, H7, H8, E5, E9, E10, E11 and C10.

**Figure 4 vaccines-08-00613-f004:**
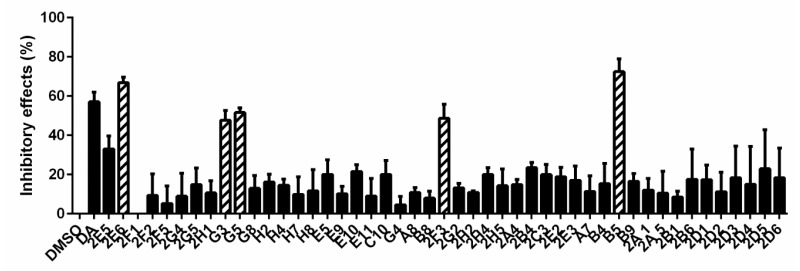
The calculated inhibitory effects of 47 natural products on *B. microti* determined by microscopy. The error bars indicate standard error of the means for each tested group. Five drugs (2E6, G3, G5, 2F3, and B5) with significant differences from DMSO were selected as potential drugs against *B. microti*. The drugs were in the sequence of DMSO, DA, 2E5, 2E6, 2F1, 2F2, 2F5, 2G4, 2G5, 2H1, G3, G5, G8, H2, H4, H7, H8, E5, E9, E10, E11, C10, G4, A8, B8, 2F3, 2G2, 2H2, 2H4, 2H5, 2A4, 2B4, 2C3, 2E2, 2E3, A7, B4, B5, B9, 2A1, 2A5, 2B1, 2B6, 2D1, 2D2, 2D3, 2D4, 2D5, and 2D6.

**Figure 5 vaccines-08-00613-f005:**
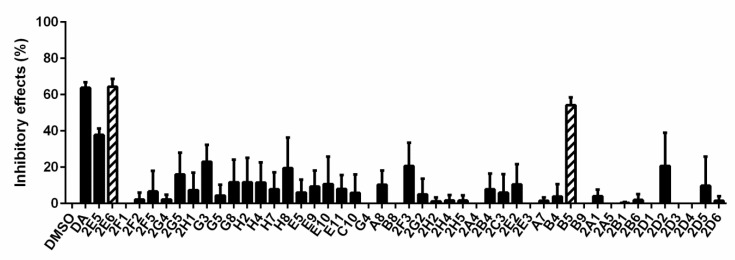
The inhibitory effects of 47 natural products on *B. microti* determined by flow cytometry. Two drugs (2E6 and B5) with significant differences from DMSO were finally selected as potential drugs against *B. microti*. The error bars indicate standard error of the means for each tested group.

**Figure 6 vaccines-08-00613-f006:**
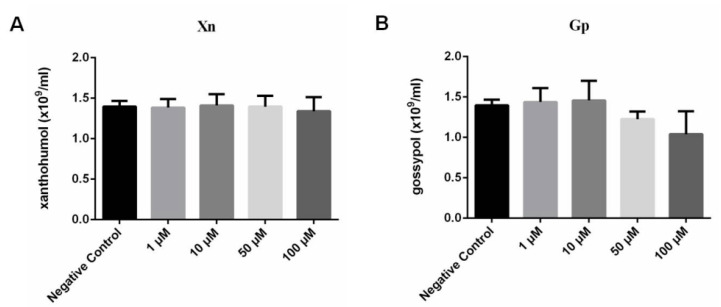
The cytotoxicity assay of xanthohumol (**A**) and gossypol (**B**). The negative control was treated with medium without drugs.

**Figure 7 vaccines-08-00613-f007:**
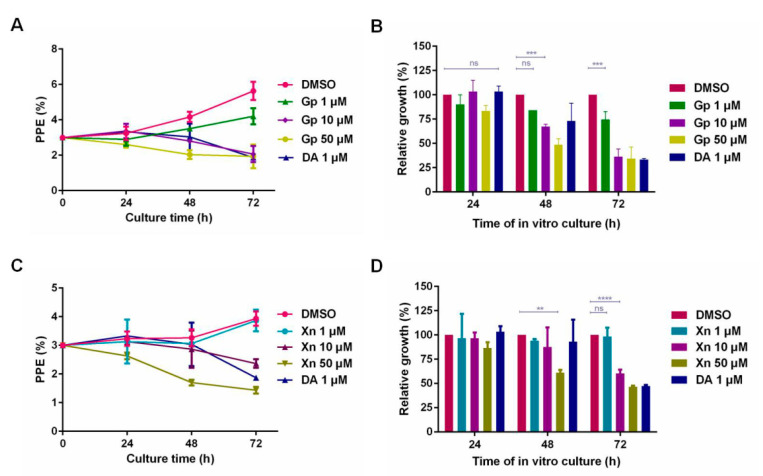
The inhibitory effect of gossypol and xanthohumol on the in vitro culture of *B. microti*. The parasitemia for treatment with different concentrations of gossypol (**A**,**B**) as well as different concentrations of xanthohumol (**C**,**D**). (“ns”: no significance, **** *p* < 0.0001, *** *p* < 0.001 and ** *p* < 0.01).

**Figure 8 vaccines-08-00613-f008:**
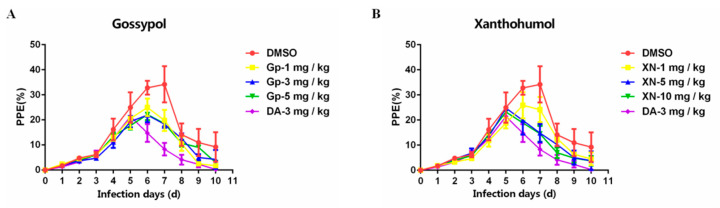
The inhibitory effect of Gp (**A**) and Xn (**B**) on the in vivo culture of *B. microti*.

**Table 1 vaccines-08-00613-t001:** The summary of cytotoxicity of gossypol and xanthohumol on different cell lines.

Nature Products	Organism/Cell Type	IC_50_	Selectivity Index ^a^
**Gossypol**	erythrocyte of KM	>100 μm	>11.8
	chronic lymphocytic leukemia cells	30 μm	3.50
	epithelial cervix cancer cells (HeLa)	31.3 μm	3.70
	brain malignant glioma cells (U87)	59.6 μm	7.00
	gastric cancer cells (M85)	39.7 μm	4.70
**Xanthohumol**	erythrocyte of KM	>100 μm	>4.7
	human colon adenocarcinoma cell lines (HT29)	50.2 μm	2.30
	human colon adenocarcinoma cell lines (HCT116)	40.8 μm	1.90
	human hepatocellular carcinoma cell lines (Huh7)	37.2 μm	1.70
	hypotriploid alveolar basal epithelial cells (A549)	30.5 μm	1.40
	epithelial cervix cancer cells (HeLa)	40.4 μm	1.90
	human hepatocellular carcinoma cell lines (HepG2)	35 μm	1.60

^a^ Ratio of the IC_50_ on cell lines to the IC_50_ of each drug on the in vitro culture of *Babesia microti*. KM: Kunming mice.

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
