# Peer review of "Xanthohumol and Gossypol Are Promising Inhibitors against Babesia microti by In Vitro Culture via High-Throughput Screening of 133 Natural Products"

_vaccines, 2020, doi:10.3390/vaccines8040613_

Round 1

Reviewer 1 Report

In the present study, the authors screened the inhibitory efficacies of 133 natural products against the in vitro growth of Babesia microti. Two compounds including Gossypol and xanthohumol exhibited the the highest anti B. microti inhibitory effect with IC50 8.47 μM and 21.40 μM, respectively. Indeed, gossypol was screened against the in vitro growth of one Babesia parasite, B, bovis. Although the presented data are interesting, the MS restricted only on the in vitro study with neglecting the inhibitory efficacies of the obtained two potent compounds (Xn and Gp ) against the in vivo growth of B. microti in mice. Taken together, although the authors mentioned the importance of combination therapy in overcoming the drug resistance and toxicity of the currently used anti- Babesia chemical compounds using natural compounds, they didn’t evaluate the inhibitory effects of Xn and Gp when used in combination with the commonly used antibabesial compounds, DA. Subsequently, in my opinion the presented data are preliminary to be published in Vacines and to use in the field. Therefore, the authors SHOULD add another experiments in vitro (evaluate the inhibitory effects of Xn and Gp when used in combination with the commonly used antibabesial compounds, DA ) and in vivo (Xn abnd Gp when administrated as monotherapy and/or in combination therapy with DA in case of in vitro experiments showed synergistic or additive interaction between 2- drugs).

Line 22, replace in this study with in this regard

Line 22,  In this study, 133 natural products were performed to preliminary screen; screened against what?

Line 136, remove “.”

Line 138, cited the used natural compounds using supplementary data (Table S1)

Line 140, add “comma“  before “respectively“

Line 141 and line 157, the authors already mentioned the abbreviations of the drugs at the introduction. No need to repeat the full name of the drug. Pls check here and in all the text

Line 151, add the version number, country and/ or state of the used software

FIGURE 1, the authors indicated that the values are presented as mean and SD, but I can’t see the SD on the figure. Also, the figures are overcrowded and not clear. I suggest moving this preliminary screening to the supplementary data or presenting these data in table to be clearer for the readers.

 Figure 2, it’s better to present also in table

Figure 3, show the inhibitory effects of DA, 2E6, G3, G5, 2F3 and B5 together with their chemical structure and formula. The readers may need to know more information about these promising compounds. Same as in figurer 1, I can’t see the SD in this figure

Figures 4 and 6, its difficult to read it. Please change to table

Lines 132, 137, change “B. microti“ to etalic. Pls check in all the text

Line 237, Pls delete “The summary of cytotoxicity results would be detailed in the discussion section“.

Please add the selectivity indices of the compounds with potent anti-B. microti (Xn and Gp ) according to the obtained IC50s and CC50s

Lines 261,262, delete the full name of drugs

Line 264, I noticed the authors used more and more here and also some times in the introduction. It’s better to use another expression. Same for All in all at line 340

In the discussion, pls suggest the name of drug compounds those have the same target of Xn and Gp and can be used in the future for the combination therapy with these two compounds against the growth of Babesia microti.

Author Response

In the present study, the authors screened the inhibitory efficacies of 133 natural products against the in vitro growth of Babesia microti. Two compounds including Gossypol and xanthohumol exhibited the the highest anti B. microti inhibitory effect with IC50 8.47 μM and 21.40 μM, respectively. Indeed, gossypol was screened against the in vitro growth of one Babesia parasite, B, bovis. Although the presented data are interesting, the MS restricted only on the in vitro study with neglecting the inhibitory efficacies of the obtained two potent compounds (Xn and Gp) against the in vivo growth of B. microti in mice. Taken together, although the authors mentioned the importance of combination therapy in overcoming the drug resistance and toxicity of the currently used anti- Babesia chemical compounds using natural compounds, they didn’t evaluate the inhibitory effects of Xn and Gp when used in combination with the commonly used antibabesial compounds, DA. Subsequently, in my opinion the presented data are preliminary to be published in Vaccines and to use in the field. Therefore, the authors SHOULD add another experiments in vitro (evaluate the inhibitory effects of Xn and Gp when used in combination with the commonly used antibabesial compounds, DA) and in vivo (Xn abnd Gp when administrated as monotherapy and/or in combination therapy with DA in case of in vitro experiments showed synergistic or additive interaction between 2- drugs).

Response:

Thank you for your valuable suggestions on how to revise this manuscript. Based on your suggestions, we revised our manuscript carefully. Firstly, even though B. microti and B. bovis both belong to Babesia genus, B. microti can lead to human babesiosis and has been a threat to public health. To date, almost all the recommended drugs against B. microti used in humans have side effects. Therefore, it is imperative to develop new drugs to inhibit B. microti. Although gossypol has been tested in B. bovis, the same drug varied in its effects on different Babesia species, suggesting that it is necessary to test the effect of this drug on B. microti. Moreover, gossypol and xanthohumol were the two drugs with better inhibitory effects against B. microti based on screening results from 133 natural drugs, indicating the necessity to further investigate them as the potential drugs against B. microti. Secondly, we have added the combination test of gossypol and xanthohumol via in vitro culture of B. microti. The results showed that gossypol and xanthohumol had antagonistic effects on B. microti, suggesting that monotherapy is better than combination therapy. DA is a first-line agent for treatment of babesiosis, but it cannot eliminate Babesia, resulting in relapse and the emergence of more and more drug-resistant isolates, thus it is now used as a typical positive control for in vitro testing assays in Babesia as indicated in the references (Sen Wang, 2020; Long Yu, 2019). Therefore, through comprehensive analysis, we think that DA may not be suitable for combination test with gossypol or xanthohumol and can only be used as a positive control in our assays. Thirdly, we added the in vivo assays of gossypol and xanthohumol against B. microti. The results showed that parasitemia significantly decreased from 24.77% to 19.87% after treatment with 5 mg/kg Xn, with a decrease rate of 39.48%. We did not do the combination test on in vivo assay because they showed antagonism on B. microti in the in vitro assays. All the experiments we added were based on the suggestions and included in the revised manuscript. We hope that the revised manuscript can meet your requirements. Thank you for your understanding.

Question 1:Line 22, replace in this study with in this regard

Response: Thank you for your suggestion. Because the manuscript has been revised, the line numbers may be different with the original one. Due to "in this regard" has been used in the last sentence, "in this study" was replaced with "here" in the revised manuscript as follows:

    Currently, the regimen of atovaquone and azithromycin is considered as the standard therapy for treating human babesiosis, which, however, may result in drug resistance and relapse, suggesting the necessity of developing new drugs to control B. microti. In this regard, natural products are promising candidates for drug design against B. microti, due to their active therapeutic efficacy, less toxicity and fewer adverse reactions to host. Here, the potential inhibitors against B. microti were preliminarily screened from 133 natural products, and 47 of them were selected for further screening.

Question 2:Line 22, In this study, 133 natural products were performed to preliminary screen; screened against what?

Response: The sentence in line 22 has been revised in the manuscript as follows:

    Here, the potential inhibitors against B. microti were preliminarily screened from 133 natural products, and 47 of them were selected for further screening.

Question 3:Line 136, remove “.”

Response: "." has been removed in line 136.

Question 4:Line 138, cited the used natural compounds using supplementary data (Table S1).

Response: Thank you for your suggestion. The 133 natural products have been listed in Table S1 and mentioned in the revised manuscript as follows:

    A total of 133 natural products (different herbal extracts) obtained from the small molecule natural product library (Selleck Chemicals, Houston, TX, USA) were used for screening potential drugs against B. microti by in vitro culture. The general information of the 133 natural products was shown in Table S1.

Question 5:Line 140, add “comma“  before “respectively“

Response: The "comma" has been added before "respectively".

Question 6:Line 141 and line 157, the authors already mentioned the abbreviations of the drugs at the introduction. No need to repeat the full name of the drug. Pls check here and in all the text

Response: Thank you for your suggestion. The full name of the drugs in line 141 and line 157 have been replaced with abbreviations. We have checked that throughout the text in the revised manuscript.

Question 7:Line 151, add the version number, country and/ or state of the used software.

Response: Thank you for your suggestion. The version number, country and/ or state of the used software has been added in the revised manuscript as follows:

    The data were analyzed by GraphPad Prism (version 6.0, La Jolla, CA, US).

Question 8:FIGURE 1, the authors indicated that the values are presented as mean and SD, but I can’t see the SD on the figure. Also, the figures are overcrowded and not clear. I suggest moving this preliminary screening to the supplementary data or presenting these data in table to be clearer for the readers.

Response: Thank you for your comment. During data analysis by GraphPad Prism 6.0, the mean and SD were used in figure generation, but they failed to appear in the figures, thus we have deleted the sentence " Each value indicated the mean ± standard deviation for three wells" in the figure legend. While figure 1 was overcrowded, it may facilitate the observation of differences. Meantime, the data presented in figure 1 were added in Table S2 for clarity.

Question 9:Figure 2, it’s better to present also in table

Response: Thank you for your comment. It is better to keep figure 2 because it was easier to observe the differences. Meantime, the data presented in figure 2 were also shown in Table S2 for clarity.

Question 10:Figure 3, show the inhibitory effects of DA, 2E6, G3, G5, 2F3 and B5 together with their chemical structure and formula. The readers may need to know more information about these promising compounds. Same as in figurer 1, I can’t see the SD in this figure.

Response: Thank you for your comment. The chemical structure and formula of the five natural products obtained from the small molecular natural product library software (Selleck Chemicals, https://www.selleckchem.com/) were summarized in Figure S1. Meanwhile, the information of G3, G5, and 2F3 has been shown in detail in the revised manuscript (discussion section) as follows:

    The five natural products were 2F3 (palmatine chloride), B5 (Gp), 2E6 (Xn), G3 (cryptotanshinone) and G5 (honokiol). Palmatine chloride as a hydrochloride salt of palmatine has strong anti-malarial activity with low cytotoxicity and is also a valuable Alzheimer's disease modifying strategy (3,4). Cryptotanshinone, a quinoid diterpene isolated from Salvia miltiorrhizabunge showed various anti-cancer activities (5). Honokiol originally obtained from magnolia extract is reported to have inhibitory activity against multiple malignancies (6). However, palmatine chloride, cryptotanshinone and honokiol were subsequently removed through determination of parasitemia by flow cytometry. Gp and Xn were finally identified to significantly inhibit the growth of the B. microti by the combined determination of microscopy and flow cytometry.

    During data analysis by GraphPad Prism 6.0, the mean and SD were used in figure generation, but they failed to appear in the figures, thus we have deleted the sentence " Each value indicated the mean ± standard deviation for three wells" in the figure legend.

    Figure S1 as follows:

Question 11:Figures 4 and 6, its difficult to read it. Please change to table

Response: For the clarity of Figure 4 and 6 with a focus on the five natural products for further screening, the five natural products were shown in hollow cylinder and the other natural products are presented in solid cylinder. The sequence of the drugs was detailed in the figure legend as follows:

    The calculated inhibitory effects of 47 natural products on B. microti determined by microscopy. The error bars indicated standard error of the means for each tested group. Five drugs (2E6, G3, G5, 2F3 and B5) with significant differences from DMSO were selected as potential drugs against B. microti. The drugs were in the sequence of DMSO, DA, 2E5, 2E6, 2F1, 2F2, 2F5, 2G4, 2G5, 2H1, G3, G5, G8, H2, H4, H7, H8, E5, E9, E10, E11, C10, G4, A8, B8, 2F3, 2G2, 2H2, 2H4, 2H5, 2A4, 2B4, 2C3, 2E2, 2E3, A7, B4, B5, B9, 2A1, 2A5, 2B1, 2B6, 2D1, 2D2, 2D3, 2D4, 2D5, and 2D6.

Question 12:Lines 132, 137, change “B. microti“ to etalic. Pls check in all the text

Response: Thank you for your comment. All the "B. microti" in the manuscript has been checked in format.

Question 13:Line 237, Pls delete “The summary of cytotoxicity results would be detailed in the discussion section“.

Response: The sentence "The summary of cytotoxicity results would be detailed in the discussion section" in line 330 had been deleted in the revised manuscript.

Question 14:Please add the selectivity indices of the compounds with potent anti-B. microti (Xn and Gp ) according to the obtained IC50s and CC50s

Response:

Thank you for your suggestion. The selectivity indices of Gp and Xn have been added in Table 1 as follows:

Table 1. The summary of cytotoxicity of gossypol and xanthohumol on different cell lines.

Organism/cell Type

IC50

Selectivity Indexa

Gossypol

erythrocyte of KM

>100 μM

>11.8

chronic lymphocytic leukemia cells

30 μM

3.50

epithelial cervix cancer cells (HeLa)

31.3 μM

3.70

brain malignant glioma cells (U87)

59.6 μM

7.00

gastric cancer cells (M85)

39.7 μM

4.70

Xanthohumol

erythrocyte of KM

>100 μM

>4.7

human colon adenocarcinoma cell lines (HT29)

50.2 μM

2.30

human colon adenocarcinoma cell lines (HCT116)

40.8 μM

1.90

human hepatocellular carcinoma cell lines (Huh7)

37.2 μM

1.70

hypotriploid alveolar basal epithelial cells (A549)

30.5 μM

1.40

epithelial cervix cancer cells (HeLa)

40.4 μM

1.90

human hepatocellular carcinoma cell lines (HepG2)

35 μM

1.60

aRatio of the IC50 on cell lines to the IC50 of each drug on the in vitro culture of Babesia microti.

Question 15:Lines 261,262, delete the full name of drugs

Response:

The full names of the drugs have been deleted in the revised manuscript as follows:

    To date, a series of drugs are available such as ID and DA, which are two effective drugs widely used in treatment of animal babesiosis.

Question 16:Line 264, I noticed the authors used more and more here and also some times in the introduction. It’s better to use another expression. Same for All in all at line 340

Response: Thank you for your suggestion. " more and more " and " All in all " have been changed in the revised manuscript for clarity and brevity.

Question 17:In the discussion, pls suggest the name of drug compounds those have the same target of Xn and Gp and can be used in the future for the combination therapy with these two compounds against the growth of Babesia microti.

Response:

Thank you for your suggestion. The related information has been added in the revised manuscript as follows:

    In this article, the combination of Gp and Xn for in vitro assay showed that they have an antagonistic effect on B. microti, indicating monotherapy is better than combination therapy. For the combination therapy strategy, those with a similar drug target will be more suitable. Mycophenolate mofetil, AGI-5198, disulfiram, olutasidenib, and brequinar are potential dehydrogenase inhibitors, which can be further studied for combination with Gp to test the inhibition roles against B. microti. For Xn, previous studies have shown that it could inhibit cyclooxygenase 1 (COX-1) and COX-2 activity in mucosal protection. Diclofenac sodium, lornoxicam, and ketorolac tromethamine salt were selective COX inhibitors, which could be further investigated to combine with Xn against B. microti.

Reviewer 2 Report

Major comment :

In my opinion the aim of this work is more toward drug treatment not vaccination, which outside the scope of the journal because there was no any vaccine trial or any immunological work. Even in the conclusion section the authors mentioned that this study will help in “development of novel drugs against B. microti”.

Minor comment

In the introduction  

Line 36: “33 states” of what country or countries?

“regards” this word was repeated many times throughout the manuscript, please use alternatives word

In the material and methods

Line 98: what the abbreviation “P.P” stand for?

Line 136: authors should mention that the 133 list of the product are in the supplementary file as they didn’t refer to it in the manuscript.

In this section, depending on what that the authors decided to choose the 10 μM for initial concentration? Please justify this concentration because not all the natural products will have the same effect with the same concentration since they are chemically different from each other.

The same Comment regarding the DMSO.

In the result section:

Authors didn’t mention in a separate supplementary file what are the selected 47 natural products.

Regarding the figures:

Figure #1: the data is not clear at all

Figure #2, #4 and #6: the x axis needs to be clear to the reader

Figure#5: is the data in this figure is a representative sample from the 47 samples? please explain.

Figure #7: Please mention which control, negative or positive control in the figure.

Figure #8: the error bar is missing in the Gp1 concentration in the 48 hr in the figure 8B  

In the discussion:

Lines from 286-273: repetitive information since the authors mentioned that in the introduction section even with the same reference.

Author Response

Major comment:

In my opinion the aim of this work is more toward drug treatment not vaccination, which outside the scope of the journal because there was no any vaccine trial or any immunological work. Even in the conclusion section the authors mentioned that this study will help in “development of novel drugs against B. microti”.

Response: Thank you for your comment. We completely understand your concern and had similar considerations before we submitted the manuscript to this special issue. So we sent the title and abstract to managing editor and academic editor in advance to check if it is suitable for this special tissue. We received the reply from the editor that the abstract fitted well with the scope of this special issue "Advances in Malaria Vaccines". Therefore, we submitted the full paper to this special tissue. We are very grateful for your significant comment, and we will try our best to improve the experiments and manuscript to meet the requirements for publication in this journal.

Minor comment

In the introduction

Question 1: Line 36: “33 states” of what country or countries?

Response: Thank you for your comment. Because the manuscript has been revised, the line numbers may be different with the original one. The "33 states" are from the United States. We have revised the sentence in line 34 based on the references for clarity as follows:

    Facing the challenges, the Centers for Disease Control and Prevention regarded it as a nationally notifiable disease and has been monitoring it in the United States since January 2011 (1).

References are as follows:

  1. Jajosky, R. P., Jajosky, A. N., and Jajosky, P. G. (2020) The Centers for Disease Control and Prevention and State Health Departments should include Blood-Type Variables in their Babesiosis Case Reports. Transfusion and apheresis science : official journal of the World Apheresis Association : official journal of the European Society for Haemapheresis, 102824

Question 2:“regards” this word was repeated many times throughout the manuscript, please use alternatives word

Response: Thank you for your comment. The "regard" has been replaced with "treated", "considered", "used" in the revised manuscript.

In the material and methods

Question 3:Line 98: what the abbreviation “P.P” stand for?

Response: Sorry for the spelling error. The abbreviation should be “PRC” in line 98 for People's Republic of China. It has been added in the revised manuscript as follows:

    All the experimental animals were housed and treated in accordance with the stipulated rules for the regulation of the administration of affair concerning experimental animals of the People's Republic of China.

Question 4: Line 136: authors should mention that the 133 list of the product are in the supplementary file as they didn’t refer to it in the manuscript.

Response: Thank you for your comment. The 133 list of the natural products has been added in Table S1 and mentioned in line 134 of the revised manuscript.

Question 5:In this section, depending on what that the authors decided to choose the 10 μM for initial concentration? Please justify this concentration because not all the natural products will have the same effect with the same concentration since they are chemically different from each other.

Response: Thank you for your comment. To date, in screening of drugs against Babesia, only one or two drugs are contained in most tests, so a concentration gradient is used. In our study, 133 natural products were subjected to a high-throughput screening, so we had to select a moderate concentration for initial screening. The initial concentration could not be too high, otherwise most of the drugs will be involved in rescreening. Meanwhile, the initial concentration could not be too low, otherwise too few drugs can be selected for rescreening. Therefore, based on some references (Sen Wang, 2020; Long Yu, 2019), we finally decided to select 10 μM as the initial concentration. We understand your concern that the effects of natural products vary with their concentrations. However, the aim for this article was to select the drugs with the most significant efficacy rather than confirm the optimal concentration for the 133 natural products. So we screened the 133 natural products with one concentration and selected those with most significant inhibitory effects for further analysis.

References:

  1. Wang, Sen, et al. "Inhibitory effects of fosmidomycin against Babesia microti in vitro." Frontiers in Cell and Developmental Biology 8 (2020): 247.
  2. Yu, Long, et al. "Identifying the Naphthalene-Based Compound 3, 5-Dihydroxy 2-Napthoic Acid as a Novel Lead Compound for Designing Lactate Dehydrogenase-Specific Antibabesial Drug." Frontiers in Pharmacology 10 (2019).

Question 6:The same Comment regarding the DMSO.

Response: The concentration of DMSO used in this article was 0.01% because some of the natural products were prepared in DMSO, with DMSO being used as a negative control. Meanwhile, 0.01% DMSO will not influence the growth of parasites.

In the result section:

Question 7:Authors didn’t mention in a separate supplementary file what are the selected 47 natural products.

Response: Thank you for your comment. The full names of the 47 natural products were detailed in Table S1 and their code numbers were mentioned in the revised manuscript in line 187 as follows:

    A comparison with DA and DMSO further selected 47 natural products: A7, A8, B4, B5, B8, B9, C10, E6, E9, E10, E11, G3, G4, G5, G8, H2, H4, H7, H8, 2A1, 2A4, 2A5, 2B1, 2B4, 2B6, 2C3, 2D1, 2D2, 2D3, 2D4, 2D5, 2D6, 2E2, 2E3, 2E5, 2E6, 2F1, 2F2, 2F3, 2F5, 2G2, 2G4, 2G5, 2H1, 2H2, 2H4 and 2H5. The full names of the 47 selected natural products were detailed in Table S1.

Regarding the figures:

Question 8:Figure #1: the data is not clear at all

Response:

Thank you for your comment. Even though figure 1 was overcrowded, it may facilitate the observation of the differences. Meantime, the data presented in figure 1 were shown in Table S2 for clarity.

Question 9:Figure #2, #4 and #6: the x axis needs to be clear to the reader

Response:

We hope to keep figure 2. Even though figure 2 was overcrowded, it may facilitate the observation of the differences. Meantime, the data presented in figure 2 were shown in Table S2 for clarity. For clarity in figure 4 and 6 with a focus on the five natural products for rescreening, the five natural products are shown in hollow cylinder and the other natural products were presented in solid cylinder. The sequence of the drugs was detailed in the figure legend.

Question 10:Figure#5: is the data in this figure is a representative sample from the 47 samples? please explain.

Response: Thank you for your comment. Figure 5 showed the results of RBCs treated with 25 μM xanthohumol (one of the two most effective natural products) as a representation of the flow cytometry results. As figure 5 can only show a part of the cytometry results not all the data, so we deleted it in the revised manuscript.

Question 11:Figure #7: Please mention which control, negative or positive control in the figure.

Response: Thank you for your comment. The negative control in Figure 7 was the erythrocytes treated only with medium without drugs. There was no positive control because the indication for this cytotoxicity assay was erythrocyte lysis, which can be more easily observed and analyzed without a positive control.

Question 12:Figure #8: the error bar is missing in the Gp1 concentration in the 48 hr in the figure 8B 

Response: Thank you for your comment. We have carefully checked the raw data of 1 μM of Gp in the 48 h, and the data for triplicate were all the same, which is the reason for no error bar in Gp1 concentration at 48 h in Figure 8B.

In the discussion:

Question 13:Lines from 286-273: repetitive information since the authors mentioned that in the introduction section even with the same reference.

Response:

Thank you for your comment. The similar content in the introduction section has been deleted to avoid repetition.

Reviewer 3 Report

This is a study of the in vitro antiparasite activity of several natural products most of which reported as acronyms (?). I think that the Authors should give more details about these products, or refer only to xanthohumol and gossypol (the effective ones). The manuscript should be edited by a native English speaker

other concerns are reported below.

line 16 - please, omit (B. microti)

line 20 -their active..

line 23 - confirmed by microscopy, please rewrite the sentence more properly

lines 33-35 - the sentence is not correctly written

line 37 - kinds of Babesia do not exist, please rewrite "more than one hundred species of Babesia have been reported"

line 38 - omit (B. microti)

line 40 - and is transmitted ...by transfusion and..

line 42 - show increasing..

line 43 - please, omit infection. Infections are usually asymptomatic

line 46 - neonates

line 49 - rewrite the sentence

line 58 - showed instead of have, failures instead of fails

line 60 - an analogue

line 63 - in B. microti infected

line 74 - their

line 87-94 -please move the sentence at line 72 and add "the aim of the present study was ..."

line 103 - original?

line 109 - was microscopically determined on blood smears

PSG and PSG+G per extenso at the first mention

lines 118, 245 and 164 - microti

line 136 - 133 different herbal extracts...

line 138 - in vitro in italics

line 149 - omit the

lines 166, 347 and 190 - delete kinds of

line 269 and following - please rewrite the whole sentence

Author Response

This is a study of the in vitro antiparasite activity of several natural products most of which reported as acronyms (?). I think that the Authors should give more details about these products, or refer only to xanthohumol and gossypol (the effective ones). The manuscript should be edited by a native English speaker.

Response:

Thank you for your comments. Based on your suggestions, the information of the 133 natural products has been added in Table S1, including molecular weight, target and other details. Meantime, in the manuscript (discussion section), we have added related information about the five natural products screened by microscopy with significant inhibitory effects on B. microti as follows:

    The five natural products were 2F3 (palmatine chloride), B5 (Xn), 2E6 (Gp), G3 (cryptotanshinone) and G5 (honokiol). Palmatine chloride as a hydrochloride salt of palmatine has strong anti-malarial activity with low cytotoxicity and is also a valuable Alzheimer's disease modifying strategy (3,4). Cryptotanshinone, a quinoid diterpene isolated from Salvia miltiorrhizabunge showed various anti-cancer activities (5). Honokiol originally obtained from magnolia extract is reported to have inhibitory activity against multiple malignancies (6). However, palmatine chloride, cryptotanshinone and honokiol were subsequently removed through the determination of parasitemia by flow cytometry. Gp and Xn were finally identified to significantly inhibit the growth of B. microti by the combined determination of microscopy and flow cytometry.

Question 1:line 16 - please, omit (B. microti).

Response: Thank you for your comment. Because the manuscript has been revised, the line numbers may be different with the original one. B. microti in line 16 has been deleted in the revised manuscript.

Question 2:line 20 -their active.

Response: Thanks for your suggestion. "its" in line 21 has been replaced with "their".

Question 3:line 23 - confirmed by microscopy, please rewrite the sentence more properly.

Response: Thank you for your suggestion. We have revised the sentence as "5 of the 47 natural products were further confirmed by microscopy" for clarity.

Question 4:lines 33-35 - the sentence is not correctly written

Response: Thank you for your comment. The sentence in line 34-36 has been revised as "Facing the challenges, the Centers for Disease Control and Prevention (CDC) regarded it as a nationally notifiable disease and has been monitoring it in the United States since 2011 ".

Question 5:line 37 - kinds of Babesia do not exist, please rewrite "more than one hundred species of Babesia have been reported"

Response: Thank you for your suggestion. The sentence in line 36 has been revised as "To date, more than 100 Babesia species have been reported; however only a few of them can infect human and cause human babesiosis".

Question 6:line 38 - omit (B. microti)

Response: Thank you for your suggestion. The (B. microti) in line 38 has been deleted.

Question 7:line 40 - and is transmitted ...by transfusion and..

Response: Thank you for your comment. The sentence in line 38 has been revised as " B. microti is the primary agent for human babesiosis and can be transmitted by tick Ixodes scapularis, blood transfusion and placenta". The word "that" has been replaced with "and".

Question 8:line 42 - show increasing..

Response: The sentence in line 39 has been revised as "The babesiosis cases are distributed throughout the world, especially in the United States, Asia and Europe, and show an upward trend worldwide". The word "appearing" has been replaced with "show".

Question 9:line 43 - please, omit infection. Infections are usually asymptomatic

Response: The sentence in line 41 has been revised as " The clinical manifestations are similar to those of malaria, including fever, chills, haemolysis, hemoglobinuria, splenomegaly and jaundice ". The word "infection" has been deleted in the revised manuscript.

Question 10:line 46 - neonates

Response: The word "neonatus" in line 44 has been replaced with "neonates".

Question 11:line 49 - rewrite the sentence

Response: The sentence in line 47 has been revised as "Despite some recommended therapies for Babesia available, some challenges still need to be solved".

Question 12:line 58 - showed instead of have, failures instead of fails

Response: The word "have" has been replaced with "showed" and "fails" is replaced with "failure".

Question 13:line 60 - an analogue

Response: The word "a" has been replaced with "an".

Question 14:line 63 - in B. microti infected

Response: The sentence in line 63 has been revised as "The regimen of atovaquone and azithromycin is effective not only in B. microti-infected hamsters, but also in the patients that failed with clindamycin and quinine".

Question 15:line 74 - their

Response: "its" has been replaced with "their".

Question 16:line 87-94 -please move the sentence at line 72 and add "the aim of the present study was ..."

Response: Thank you for your suggestion. The sentence in line 87-94 has been moved to line 73 and the "the aim of the present study was ..." has been added in the end of the introduction section as follows:

    Against the above background, the aim of the present study was to screen out potential and effective alternative anti-Babesia drugs from natural products and find an effective method to inhibit B. microti infection.

Question 17:line 103 - original?

Response: Thank you for your suggestion. For clarity, the sentence in line 103 "original from the American Type Culture Collection and" has been deleted and revised as follows:

  1. microti ((ATCCR PRA-99™) was provided by the National Institute of Parasitic Diseases, Chinese Center for Disease Control and Prevention (Shanghai, China).

Question 18:line 109 - was microscopically determined on blood smears

Response: The sentence in line 109 " The parasitemia of the B. microti-infected KM mice was monitored by making blood smears of the tail blood, stained with Giemsa and determined by microscopy." has been replaced with " The parasitemia of the B. microti-infected KM mice was microscopically determined on blood smears."

Question 19:PSG and PSG+G per extenso at the first mention

Response: Thank you for your comment. The full name of PSG and PSG+G have been added at their first mention as follows:

    The supernatant, especially the layer of white blood cells, was removed and the pellet was resuspended with cold Puck's saline glucose buffer (PSG) to the original volume and then gently mixed. The above step was repeated three times or more until the supernatant was clear. Next, the resulting mixture was resuspended 1:1 with Puck’s saline glucose plus extra glucose (PSG+G) and stored at 4 °C for further analysis.

Question 20:lines 118, 245 and 164 - microti

Response: Thank you for your comment. In the original manuscript submitted to the system, all the B. microti was "B. microti" rather than "B. Microti", which may due to the process of format modification. We have revised "B. microti" in the revised manuscript.

Question 21:line 136 - 133 different herbal extracts...

Response: " herbal extracts " has been revised with "different herbal extracts ".

Question 22:line 138 - in vitro in italics

Response: Thank you for your suggestion. All the in vitro has been revised as "in vitro" in the revised manuscript.

Question 23:line 149 - omit the

Response: "The" in line 149 has been deleted in the revised manuscript.

Question 24:lines 166, 347 and 190 - delete kinds of

Response: All the expression of "133 kinds of natural products" and "47 kinds of natural products" in the manuscript has been revised as "133 natural products" and "47 natural products".

Question 25:line 269 and following - please rewrite the whole sentence

Response: The sentence in line 269 has been revised as " However, under the regimen of atovaquone and azithromycin, increasing evidence indicated higher dosage, longer duration, and even treatment failures for some immnuocompromised patients ".

Round 2

Reviewer 1 Report

The authors addressed all the comments and the MS is significantly improved. 

Author Response

Thank you for your comments.

Reviewer 2 Report

Thanks for the authors for their hard work to address the reviewers comments.

Author Response

Thank you for your comments.

Reviewer 3 Report

Please check the spelling of immunocompromised (instead of immnuocompromised) throughout the text

line 82  - parasitoses should be encounterd, also. You can also add a further reference, wheter is required 8malaria is not an infectious disease, such as babesiosis)

lines 334-335 - please, rewrite more properly

line 386 -please, modify in "the in vivo results"

line 391 - please, delete B, microti from abbreviations

Author Response

Q1: Please check the spelling of immunocompromised (instead of immnuocompromised) throughout the text

Response:

Thank you for your comments. Sorry for the spelling mistakes. We have checked the spelling throughout the manuscript.

Q2: line 82  - parasitoses should be encounterd, also. You can also add a further reference, wheter is required 8malaria is not an infectious disease, such as babesiosis)

Response:

Thank you for your comments. We have revised the sentence and add two references in the revised manuscript as follows:

    To date, natural products have been extensively applied in many diseases, such as infectious diseases, immunological diseases, tumor diseases and parasitic diseases(20,25-27).

The two further references:

  1. Jiang, X., Chen, L., Zheng, Z., Chen, Y., Weng, X., Guo, Y., Li, K., Yang, T., Qu, S., Liu, H., Li, Y., and Zhu, X. (2020) Synergistic Effect of Combined Artesunate and Tetramethylpyrazine in Experimental Cerebral Malaria. ACS Infectious Diseases 6, 2400-2409
  2. Kyei-Baffour, K., Davis, D. C., Boskovic, Z., Kato, N., and Dai, M. (2020) Natural product-inspired aryl isonitriles as a new class of antimalarial compounds against drug-resistant parasites. Bioorganic & medicinal chemistry 28

Q3: lines 334-335 - please, rewrite more properly

Response:

Thank you for your comments. The sentence has been revised as follows:

    Lactate dehydrogenase (LDH), an indispensable active enzyme, was reported to be involved in glycolysis, mediating the reversible metabolism of pyruvate to lactate.

Q4: line 386 -please, modify in "the in vivo results"

Response:

Thank you for your comments. "in vivo culture" in line 386 has been replaced with "in vivo results".

line 391 - please, delete B, microti from abbreviations

Response:

Thank you for your comments. B, microti has been deleted from the abbreviations.
